# Socio-Hydrological Modelling to Assess Reliability of an Urban Water System under Formal-Informal Supply Dynamics

**Rakhshinda Bano** [1,2,*] **, Mehdi Khiadani** [1] **and Steven Burian** [3]

1   School of Engineering, Edith Cowan University, 270 Joondalup Drive, Joondalup, Perth, WA 6027, Australia;
    m.khiadani@ecu.edu.au
2   US Pakistan Centre for Advanced Studies in Water, Mehran University of Engineering and Technology,
    Jamshoro, Sindh 76062, Pakistan
3   Department of Civil and Environmental Engineering, University of Utah, Salt Lake City, UT 84112, USA;
    steve.burian@utah.edu
*   Correspondence: r.bano@ecu.edu.au; Tel.: +614-26-98-64-68

**Abstract:** Increasing water scarcity in developing world cities combined with poor performance of water supply systems has led to an increasing reliance on informal water supply systems. Although the availability of informal supply provides a coping mechanism that enables water consumers to be resilient to failures in water supply, the longer-term effects on formal water supply systems (FWSS) are uncertain, with a potential reduction of tariff recovery ($R_T$), and in turn a service provider's financial sustainability. This motivates an analysis of the coevolving dynamics and feedbacks involved in water systems where formal and informal components co-exist. Investigating Hyderabad, Pakistan as a case study, a dynamic socio-hydrologic system model is built, comprised of a formal system's water and fund balance, consumer behaviour and infrastructure conditions. Simulations are executed on a monthly basis at a household level and for a 100-year period (2007–2107) using data available from years 2007–2017. Demand shift to informal is observed to be weakly associated with lower recovery rates, with household income as a major predictor. The FWSS's financial balance, predominantly driven by infrastructure condition, appears to be less sensitive to recovery of a tariff to generate sufficient revenue.

**Keywords:** formal and informal water supply systems; socio-hydrological modelling; system dynamics; consumer behaviour; tariff recovery; financial sustainability

## 1. Introduction

Informal water supply systems (INFWSSs), hereby referred to as non-piped or non-networked water supply managed by private suppliers [1], have emerged as a common adaptation strategy to increasing domestic water scarcity, especially in developing countries [2–4]. Formal water supply systems (FWSSs), i.e., piped water supply managed by water utilities, are failing to meet consumer demands for the reliable service of high-quality water [5]. This has given rise to increasing mistrust from consumers, contributing to lower water tariff recovery rates ($R_T$) of FWSSs [6]. While informal water supply has emerged as a coping mechanism enabling users to deal with uncertain water supply [7], it has also contributed concomitant negative feedbacks to the financial sustainability of FWSSs, which further feeds back in to a vicious cycle of poor performance in formal water system [8]. This suggests an analysis of the dynamics and feedbacks involved in a system where formal and INFWSSs co-exist, to support advancing urban water sustainability [9].

Financial sustainability has remained a constant challenge for all urban water utilities. Recently, the challenge has intensified as the stock of infrastructure assets globally has reached its designed service life, and financial constraints are inhibiting replacement [10]. With ageing infrastructure (and associated reduced energy efficiency and *water losses* from leakage) combined with funding shortfalls, water utilities responsible for public water supply in cities are increasingly unable to reliably meet present water demands, and future urban growth is anticipated to further stress the systems [11,12]. Financial constraints, especially in developing countries, are primarily driven by water tariffs being set below sustainable levels (necessary to cover maintenance costs of water infrastructure) and low $R_T$, leading to a lack of cost recovery [10]. A key breakdown within the system is the inability of policymakers to update water tariffs because of consumers' low willingness to pay [13]. Consumers may not be willing to pay water tariffs due to low performance of the utilities [6]. The situation triggers further processes, such as increased payments, to cope with unreliable water supply [3]. This affects consumer behaviour, which constrains future efforts to solve problems, aggravating water security issues in the long run.

The literature on socio-hydrology has enabled us to have a better understanding of coupled human–water systems [14]. Unlike traditional hydrology, socio-hydrology allows human actions to be part of water cycle dynamics to understand the coevolution of coupled human–water systems [15]. A number of studies have been recently conducted to understand this coevolution, incorporating both environmental processes and human behaviour [16]. Characterization and quantification of these interactions can help effectively manage, mitigate and adapt well to future conditions [17]. The role of INFWSS in defining FWSS's financial balance has received limited attention in the socio-hydrological literature. While the authors of [8,18,19] mainly focus on infrastructure conditions, adaptive consumer behaviour in terms of reliance on INFWSSs may be a more important factor in the developing world affecting consumer behaviour and defining financial balance. These types of feedbacks are important in modelling "co-evolutionary system models" to provide a way to study system management over a long term [17].

Since the number and extent of the INFWSSs are growing in the urbanized developing world [1,4,20], it is important to consider the effect on the overall water supply system performance in a city. Informal supply systems have been found to be more reliable sources of supply in terms of timely delivery and quality standards [21,22]. On one hand, the dismissal of the informal system's role by planners and designers has been associated with persistence in western planning ideals [23]. Informal water supply has also been questioned as the long-term sustainable solution because of higher per capita cost, challenges of regulation, and lack of equity [9]. Generally, the poorest populations are affected the most, with major proportions of their income spent on water. As informal systems increase, achieving "access to safe and affordable drinking water for all" gets farther from reality [24].

The present study endorses the relevance of the INFWSS and embraces its role in municipal water security [21,22]. The study further explores how reliance on an informal water system creates increasingly complex water supply dynamics in arid urban areas and the consequential effect on $R_T$ and the FWSS performance. This study's novelty lies in exploring feedback processes associated with the emergence of INFWSSs as a human adaptation approach to water scarcity at the household (HH) level, and the response in formal water system's reliability in terms of its ability to meet domestic (D) and non-domestic (ND) demand (which include public and private sector institutions, schools, hospitals, pump stations, small-level home industries) and its financial balance. This shift in water demand to informal supply represents human response to water scarcity. This shift may affect consumer behaviour in terms of tariff payment to a formal system which may in turn affect financial sustainability and performance, and thus in cyclic feedback, consumer wellbeing and ability or willingness to pay tariffs. While multiple studies [9,22] have discussed the possible impact of INFWSS in terms of cost incurred to the consumer, the present study presents a dynamic view of the impact an informal water system in terms of water security at the HH level as well as a formal system's fund balance.

Using Stella Architect (© isee systems inc, Lebanon, NH, USA), a system dynamics (SD) modelling tool, the relationships in coupled formal–informal systems are identified using Hyderabad Pakistan as a case study. The paper identifies interlinkages and feedback loops between formal and informal systems in the city. Stocks and flows are used to model the dynamicity of the formal water system's reliability over time and to identify potential system parameters that could affect the formal system's reliability. The system's reliability is measured in terms of household water security, defined as the combined volume or total volume of water when all sources of supply are pooled [25] and formal system's fund balance. Sensitivity analysis is employed to assess possible trajectories of output parameters, as mentioned above, under varying input variable conditions.

## 2. Modelling Approach

System dynamics modelling involves use of feedback loops or Causal Loop Diagrams (CLDs), Stocks and Flows and Time delay functions to model the nonlinear behaviour of complex systems [26].

### 2.1. Feedback Loops

Causal Loop Diagrams demonstrate the interconnection of system components. They are categorized into (1) Reinforcing feedback/positive feedback and (2) Balancing feedback/negative feedback. Reinforcing processes refer to the connections in which change in one component/variable triggers further change in the same direction, causing additional accumulation in the component which triggered the process. Balancing feedback or negative feedback, on the other hand, serves the purpose of balancing change in the system by resisting or negating change resulting from reinforced feedbacks. Feedback loops are distinctive, with interconnected circular relationships where something affects something else and in turn is affected by it. In contrast, linear cause and effect relationships comprise of a series of unidirectional cause and effect relationships.

Figure 1 frames the scope of the SD model covered in this research, highlighting feedback processes implemented in the model. This represents formal and informal water supply dynamics for the city of Hyderabad. These processes are generalizable to other urban areas in Pakistan, including Faisalabad [27], and secondary cities in the lower Indus basin including Sukkur, Thatta and Badin [28], which rely solely on surface water (i.e., Indus River and Chenab River), due to the poor quality of groundwater. R1, R2, and R3 represent reinforcing processes and B1 and B2 refer to balancing processes or negative loops. The positive and negative signs of each arrow represent link polarity, where a positive sign shows that if one parameter increases/decreases the parameter at the tail also increases/decreases. The negative sign shows that if one parameter increases the parameter at the tail decreases.

The feedback loop R1 shows how poor service of formal systems reinforce poor financial capacity feeding back to degraded service [8]. Poor service in terms of quantity or quality leads to lower consumer satisfaction with the formal system, thus a lower $R_T$, which makes up a major proportion of the formal system's annual revenue. Thus, lower recovery rates mean less cash available for maintenance, given that there are no other funding sources available.

Lower consumer satisfaction with the formal supply system due to its poor service also determines shift to the informal supply system, creating the R2 given in Figure 1. The HH surveys conducted by Imad [13] and Zaidi [29] in Hyderabad city have revealed higher resident dissatisfaction with the FWSS, as 60% of respondents (N = 360) reported high dissatisfaction with the system and, thus, an increased reliance on the INFWSS. R2 highlights this shift in demand, further increasing the HHs' total cost of water supply as the INFWSS charges higher. Imad [13] noted per litre charges to be nine times higher in the INFWSS in Hyderabad city. Increased expenditure on water supply due to reliance on INFWSS may lower consumers' willingness to pay tariffs to the FWSS, which may further lead to low revenue and poor service condition of the FWSS, as noted in R1.

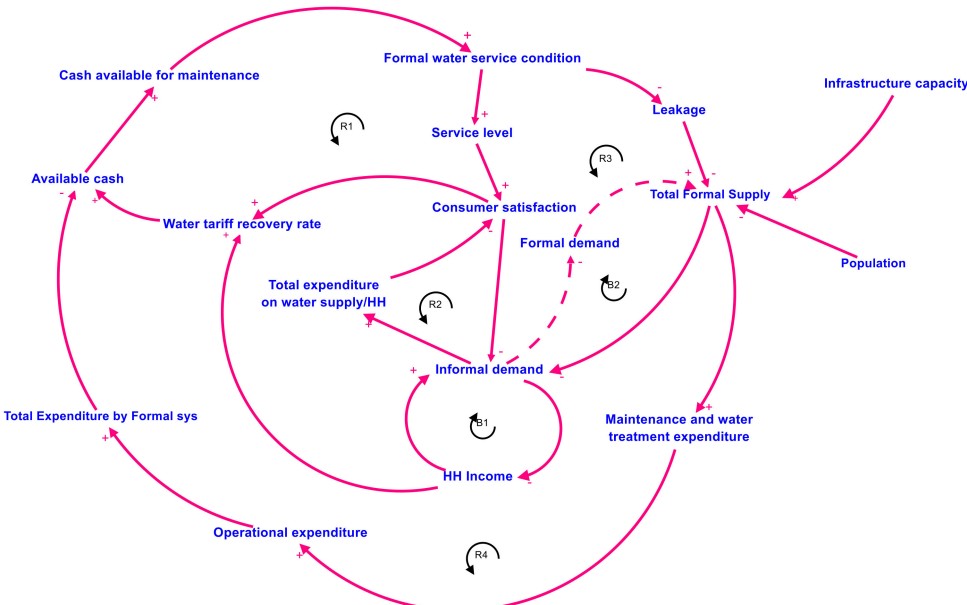

**Figure 1.** Feedback processes in formal–informal supply dynamics.

R3 portrays the impact of decreasing formal supply associated with infrastructure degradation over time, due to lack of maintenance and rehabilitation. Poor condition is thus associated with higher *water system losses* and a decrease in water availability to consumers [8]. To meet this supply–demand gap HH demand shifts to informal [30], feeding back to increases in household expenditure and decreases in $R_T$, which reduces FWSS revenue. R4 presents the increase in maintenance and water treatment expenditure associated with ageing infrastructure and related *water losses*. Increases in maintenance expenditure and water treatment lower the FWSS's fund balance, reducing available cash needed for maintaining the service condition [18]. Population and infrastructure capacity are modelled as stressors that affect total supply and determine treatment and operational expenditure and thus, the formal system's performance.

Balancing loop 1 (B1) highlights the role of HH income in defining demand for the INFWSS. The FWSS's poor performance or low reliability in terms of quality of water supply, makes the water being supplied as an inferior good. The higher the HH income, the higher the HH's affordability to have access to informal water supply. B2 presents the balancing feedback of increase in informal demand and resulting decrease in burden on the formal system to meet the demand. This feedback process is, however, out of the scope of this study and is presented to show all forms of the possible feedbacks that may exist.

*2.2. System Components and Model Building*

Using STELLA Architect, urban water system components are defined in four different modules or sectors (Figures S1–S5, provided in Supplementary Material). Each one presents a certain component that functions independently, where later these modules are integrated to model the system as a whole.

Stocks represent accumulations in the system, which can either be concrete or abstract. Flows represent the processes that bring change in stocks/accumulations (Equation (1)). In our system, urban water balance, FWSS fund balance and population are represented with stocks, while variables like percent change in supply and demand, revenues and expenditures and growth rates, are represented by flows.

$$Stock(t) = \int_{t0}^{t} [Inflow(t) - Outflow(t)]dt + Stock(t0) \tag{1}$$

### 2.2.1. Formal System Water Balance Sector

Formal system water balance sector models total formal supply and demand. Total supply refers to the quantity of piped water available to consumers through the urban water utility, i.e., Water and Sanitation Agency (WASA), in this case of Hyderabad city, categorized into domestic ($S_D$) and non-domestic ($S_{ND}$) (Equation (2)), with a constant proportion of 55 to 45 throughout the simulation period. Water for Hyderabad city is supplied from the Indus River directly pumped at different locations upstream and downstream of Kotri Barrage. Water is initially stored in 3.1 MCM (700 MG) storage lagoons and diverted to treatment plants located at different locations with the total storage capacity of 0.23 MCM (60 MGD). The model, however, simulates the initial storage or capacity of the filter plants as a single reservoir with a cumulative capacity of 0.23 MCM. Future increases in storage capacity of 0.09 MCM (25 MGD) and 0.1 MCM (26 MGD) are incorporated for the years 2023 and 2050, respectively, as per the future development plans of Planning and Development division [31]. Groundwater, on the other hand, is highly brackish and thus of not potable quality or is highly expensive to be made usable. The model, therefore, merges *water losses* due to leakages and theft under a single variable *water losses* with an initial value of 30% of total supply by the FWSS for the year 2007, as reported by WASA. Water supply losses are based on infrastructure condition (as mentioned in Section 2.2.2) and varying maintenance trends simulated under sensitivity analysis (Section 2.5).

$$Total\ formal\ supply = S_D + S_{ND} \times (1 - Leakage\ rate) \tag{2}$$

Total formal water demand comprises of domestic ($D_D$) and non-domestic ($D_{ND}$) (Equation (3)). Domestic water demand is measured at the HH level based on per capita demand ($D_{cap}$) of 152 litres per capita per day (L/c/d) as reported by WASA Hyderabad. Demand modelling is based on population growth, (keeping $D_{cap}$ constant) and consumer behaviour as explained in Section 2.2.3. Non-domestic consumers include all public and private sector institutions, schools, hospitals, pump stations and small level home industries. An aggregate value for the category obtained from WASA for the year 2017 has been used to represent the cumulative ($D_{ND}$), which grows at an annual rate ($GD_{ND}$) of 1.5% [32]. The values for years 2007–2016 were obtained backcasting the value of 2017 linearly using $GD_{ND}$.

$$Total\ water\ Demand = \left(D_{cap} \times Total\ Population\right) + (D_{ND} \times GR_{ND}) \tag{3}$$

### 2.2.2. Infrastructure Condition Sector

Adapting from [8], the infrastructure condition sector models the infrastructure condition of the formal water system focused on watermains, as they make up 80% of the life-time costs of water supply infrastructure [18]. With a total length of 200 km, watermains have been categorized into five condition group stocks namely *Condition Group 20, Condition Group 30, Condition Group 40–60, Condition Group 60–80* and *Condition Group 80–100*. An age-based deterioration function is used to shift the infrastructure condition from one condition group to the next, allowing each condition group to stay in the group for 20 years before being moved to next. Rehan, Knight, Haas and Unger [8] have justified network age being strongly correlated to pipe condition. Since the average service life of a water supply network is usually 100 years; thus, a *Renewal* function is used in the baseline scenario to present network replacement after it ages to 100 years under the *Condition Group 80–100.* The *Renewal* function moves the pipe from Condition Group 80–100 back to Condition Group 20. Additional scenarios for infrastructure ageing have also been tested under sensitivity analysis, under which infrastructure is allowed to age beyond 100 years with no replacement until the end of the simulation period, i.e., 2107 (Figure S5a). Since watermains in Hyderabad city were laid in 1976 [28], the total length of the watermains falls under the category of *Condition Group 30* for the model start year, i.e., 2007. The infrastructure condition sector is connected to the water balance and the formal system fund balance sectors through losses associated with infrastructure ageing and inflated maintenance cost to maintain the losses (as shown in Figure S2).

### 2.2.3. Consumer Behaviour

Consumer behaviour modelling involves simulation of the demand shift from formal to informal water supply system at the HH level and its feedback to the HH water balance and the FWSS fund balance. *Total monthly informal demand* at the HH level and *Recovery rate of water fee* at the HH level are used as indicator variables of consumer behaviour.

Poor condition of the FWSS with limited treatment and storage capacity as well as losses associated with poor maintenance, increasing water demand associated with population growth have been extensively reported as major reasons for the emergence of INFWSSs in developing countries [9,33]. Thus, the demand shift to INFWSS is modelled as a residual demand [30], driven by the supply–demand gap created under increasing demand over time, as majorly driven by population and a decreasing supply associated with infrastructure degradation and the resulting losses (Equation (4)). An analysis of primary data by [13] for the year of 2016 revealed the observed informal demand to be 10 m$^3$ lower than the estimated value based on residual demand modelling at HH level. Thus, a difference of 10m$^3$ per month ($HH_{min}$) is kept constant for the simulation period to estimate informal demand growth under residual demand modelling.

$$D_{INF(SD)} = HH_{\min} - T_{B(t)} + D_{INF(t-1)} \tag{4}$$

where $D_{INF(SD)}$ is the informal demand under given supply–demand gap. $T_{B(t)}$ is the total household water balance per month, driven by population change, infrastructure condition and capacity. $D_{INF(t-1)}$ is the informal demand for the previous timestep.

According to Reynaud and Romano [34], the demand of commodities are usually determined by the consumer income as well as price and prices of substitutes or complements other than consumers preferences. Moreover, preliminary analysis of the data collected by [13] has revealed income to be a major factor driving the informal water demand in Hyderabad city. Thus, the effect of income on the demand shift is modelled separately using *income elasticity* value (Equation (5)). The price elasticity of a single HH for the formal water supply cannot be captured, as consumers pay a flat rate depending on the size of their house. However, HH survey data by Imad [13], available for different income groups in the city, can be used to measure the *income elasticity*. Three different values of *income elasticity* were obtained for lower-, middle- and higher-income groups with values of 0.98, 1.45 and 1.89, respectively. The value for the middle-income group was used in the model, as the middle-income group constituted the major proportion of the total population [13]. An average monthly income value of USD 250 (PKR 25,000) was used to model the real income growth using Equation (6), with an inflation rate of 6% [35]. Informal water demand per HH per month (9.04 m$^3$) was obtained for the year 2016–2017 by Imad [13] used as the baseline value to simulate future possible growth in informal demand for the years 2017–2107. With limited data on informal supply, it was assumed to be equal to demand [30].

$$D_{INF(I)(t)} = D_{INF(I)\ (t-1)} * Income\ elasticity * Average\ income\ change\ rate \tag{5}$$

$$Real\ Income = Nominal\ income\ /\ (1\ +\ Inflation\ Rate) \tag{6}$$

where $D_{INF(I)}$ is the demand shift under changing income driven by *income elasticity* and income change rate.

Logistic regression analysis can be used to model the impact of the demand shift on $R_T$ at the HH level using Equation (8). Household survey data collected by Imad [13] for the informal demand, monthly income and payment to the utility for a sample size of 368 was used to run logistic regression in R 4.0 software to obtain the sensitivity of the $R_T$ to income and informal demand at the HH level (details on data analysis are provided in Text S1). Equation (9) was further incorporated in the SD

model to plot the sensitivity of the $P(R_T)$ against the demand shift and income change simulated by the model.

$$\ln\left(\frac{Odds_{R_T}}{1 - Odds_{R_T}}\right) = a + b(Informal\ demand) + c(Income) \tag{7}$$

$$Odds_{RT} = \frac{1}{\left(1 + e^{-(0.27 - 0.0002(Informal\ demand) + 1.8(Income))}\right)} \tag{8}$$

$$P_{RT} = \frac{Odds_{R_T}}{(1 + Odds_{R_T})} \tag{9}$$

*Odds* $_{RT}$ here refers to the odds of HH's willingness to pay water fees to the formal system, where a is the intercept and *b* and *c* are logistic regression coefficient values obtained for the total informal demand by each HH and the average monthly HH income, respectively. The coefficient value of c represents the shift from one income group to another (e.g., lower to middle and middle to higher), where the value for a unit change in income is obtained by dividing the value with 25,000, since each income group has been categorized based on a difference of PKR 25,000. Odds are defined as the ratio of the probability of success to the probability of failure. Here, the probability of success is HH's payment to the formal system and probability of failure refers to non-payment of the tariff. The odds value can range from 0 to positive infinity. $P_{RT}$ is the probability of $R_T$ and is obtained using Equation (9).

### 2.2.4. Formal System Fund Balance

The FWSS fund balance sector is comprised of revenues and expenditures of water utility, i.e., Hyderabad WASA (Equation (10)). The *revenue* is estimated by multiplying the total water demand with monthly household water tariff (USD 3) and $R_T$ (50%) as reported by WASA for the year 2017 (Equation (11)).

$$Fund\ Balnace(t) = Fund\ balance(t - dt) + (Revenue - OpEx * dt) \tag{10}$$

$$Revenue = Total\ water\ consumption \times Household\ water\ tariff \times R_T \tag{11}$$

$$\begin{aligned} OpEx\ =\ &[Unit\ price\ OpEx\ filter\ plant \times Volume\ produced] \\ &+[Unit\ price\ OpEx\ network \times total\ length\ watermain] \\ &\times(1 + condition\ multiplierOpEx/100) \end{aligned} \tag{12}$$

The FWSS's expenditures comprised of Operational (*OpEx*) and Capital Expenditure (*CapEx*). The operational expenditure covered the water treatment and maintenance cost of the watermains (Equation (12)). *CapEx*, on the other hand, included infrastructure rehabilitation and replacement. The study, however, limited expenditure modelling to *OpEx* as *CapEx* relies majorly on government funds [28] and depends on the budget available. Moreover, the major focus of the study is analysing the impact of consumer behaviour on the FWSS's finance, thus, exclusion of the government funds from the *revenue* and *CapEx* may not affect the results of the study. Financial sustainability of the formal system also requires self-sufficiency of the system [19], with revenue mostly generated by consumers. Thus, the estimation of financial deficit excluding external funding may also help in indicating actual cost burden on consumers for maintaining a good water supply system. *OpEx* usually increases with infrastructure ageing, where a *condition multiplier* variable has been used adapting [8] to inflate the *OpEx* with infrastructure ageing and multiplied with a unit cost of maintenance (per meter of watermains) obtained from [8]. Unit price per cubic meter of water produced by filter plant (USD 0.03) is obtained from annual budget data provided by the WASA. To keep all costs simple, inflation and depreciation rates were assumed to be equal and unit costs for maintenance were kept constant for the simulation period (adapted from [8]).

### 2.3. Data and Sources

Section 2.2 provides the initial values of all the parameters being used to model interrelationships. Data sources included published and grey literature from the relevant departments of the local government such as the Water and Sanitation Agency, Bureau of Statistics and Irrigation Department which oversaw the water allocation at the province level. Moreover, non-government development organizations and research and academic institutions such as the Centre for Advanced Studies in Water at Mehran University (USPCAS-W) were a major source of information about the baseline situation of the water supply in the city.

### 2.4. Model Scope, Assumptions, Calibration and Validation

Model boundaries for the water demand included the service area (referred to as Hyderabad city in the document) of Hyderabad district in Sindh Pakistan. Individual HHs in the city were chosen as the spatial unit of analysis whereas, simulations were run on a monthly basis for the years 2007–17 as baseline scenario, followed by simulations for sensitivity analysis for the time period of 2017–2107. The baseline time period was chosen based on the availability of the data. Processes such as population growth and water demand growth were incorporated on an annual scale. The model has been designed into sectors explained in Section 2.2. Simulations in Stella Architect were based on numerical integration including the Euler's method and the Runge Kutta method, where Euler's method was chosen to run the simulations on a monthly basis. This study focuses more on output variable trends and patterns instead of predicting exact future values thus, a number of simplifications have been made for input variables.

Informal supply is assumed to be equal to demand. It is reasonable to assume them to be the same as the informal system runs on profit, whereby supply is driven by demand [30]. Irrespective of changes in river flow, the city receives its share in the model. This is supported by the fact that the city's total demand is very low against the total supply in the river at Kotri, averaging 38,723 per day (estimated for the normal years 2003–2014) [36]. Moreover, the country's water policy keeps "water for people" as the priority over other uses [37]. Thus, it is assumed that, under low flows, other users (e.g., agriculture and industry) may get a low share, which is not covered in the study. Table 1 enlists the ranges of input parameters for sensitivity analysis that may allow us to explore the possible range of trajectories of output variables. A limited number of variables have been calibrated and are mentioned below:

Demand shift to informal: There are limited data to validate the demand shift over time. Per household demand for the year 2016 is used as a starting value to predict future possible shifts associated with changing supply and demand and income change. However, news sources [38] help validate increasing demand trends and other factors such as poor performance of the formal system, which are a major reason behind this shift. Validation of results through newspaper articles is a common practice for studies focused on developing countries [39,40].

Population growth: Data for the population is calibrated (provided in Table S2) to test the model's ability to simulate demand based on population growth. The population growth rate was revised from 2 to 1.93 percent based on actual population observed during the 2017 census [41]. Population growth modelling did not incorporate any growth limit as the net growth rate already incorporates the emigration and death rates.

**Table 1.** Summary of scenario conditions for sensitivity analysis.

| Scenario | $GR_D$ (%) | $GR_{ND}$ (%) | ST (MCM/Month) | Water Tariff (USD/Month) | Tariff Recovery Rate (%) | System Losses (%) | *OpEx (Condition Multiplier)* | Change in Recovery Rate (%) |
|---|---|---|---|---|---|---|---|---|
| 1.1 | 0 | 0 | 6.8 | 3 | 0.5 | 30–40 | | 0 |
| 1.2 | 1.94 | 1.5 | t1 = 6.8, t192 = 9.8, t512 = 12.8 | 3 | 0.5 | 30–40 | | 0 |
| 1.3 | 3.6 | 1.5 | t1 = 6.8, t192 = 9.8, t512 = 12.8 | 3 | 0.5 | 30–40 | *Condition multiplier 1* | 0 |
| 2.1 | 0 | 0 | 6.8 | 3 | 0.5 | 30–40 | (Figure S5d) | 0 |
| 2.2 | 0 | 0 | 6.8 | 3 | 0.8 | 30–40 | | 0 |
| 2.3 | 0 | 0 | 6.8 | 3 | 1.0 | 30–40 | | 0 |
| 2.4 | 0 | 0 | 6.8 | 12 | 0.5 | 30–40 | | 0 |
| 2.5 | 0 | 0 | 6.8 | 12 | 0.8 | 30–40 | | 0 |
| 2.6 | 0 | 0 | 6.8 | 12 | 1.0 | 30–40 | | 0 |
| 3.1 | 0 | 0 | 6.8 | 3 | 0.5 | 30–40 | *Condition multiplier 1* (Figure S5d) | 0 |
| 3.2 | 0 | 0 | 6.8 | 3 | 0.5 | 30–80 | *Condition multiplier 2* (Figure S5d) | 0 |
| 4.1 | 0 | 0 | 6.8 | 3 | 0.5 | 30–40 | *Condition multiplier 1* (Figure S5d) | 0 |
| 4.2 | 0 | 0 | 6.8 | 3 | 0.5 | 30–40 | | −10 |

There are multiple socio-political factors that may also impact water supply system performance. This paper, however, focused only on the socio-hydrological context. The model did not model the hydrology given the nature of the study area boundary. The water available to the city is originally from the massive Indus Basin and the study area itself does not generate surface flows or groundwater used as a source in the city. Moreover, informal water market data is not available; thus, the study assumed informal water supply to be equal to demand, adopting the approach by Srinivasan [25]. Most of the discussed limitations have a negligible impact on the output, or they have been covered through sensitivity analysis.

*2.5. Sensitivity Analysis*

Sensitivity analysis component in Stella architect employs two types of variations, to simulate the uncertainty in model behaviour. The first, most commonly used for stochastic elements, allows to run the model parameters with random distribution. The second, allows to simulate the model behaviour with different initial conditions and constants [42]. This study used the second approach for the sensitivity analysis. Table 1 summarizes input variables chosen to be varied (highlighted) to identify their effect on the output variables, i.e., per capita water balance, demand shift from formal to informal and the formal system fund balance. Ranges of the parameters were obtained either from literature or supported by the preliminary statistical analysis of the survey data obtained by [13]. The input variables values were selected to have an ad-hoc distribution, with a total combination of 13 runs.

## 3. Results

Trajectories of the water balance at both the city and the HH level are displayed in Figure 2, followed by demand shift patterns from formal to informal (Figure 3) and their impact on $R_T$ (Figure 3: (a) Water demand shift to informal at HH and (b) city level under current infrastructure capacity and population increase versus income change. The formal system's fund balance was simulated to assess possible impacts of the demand shift from the formal to the informal system (Figure 4), followed by sensitivity analysis of the output parameters (Figures 5 and 6).

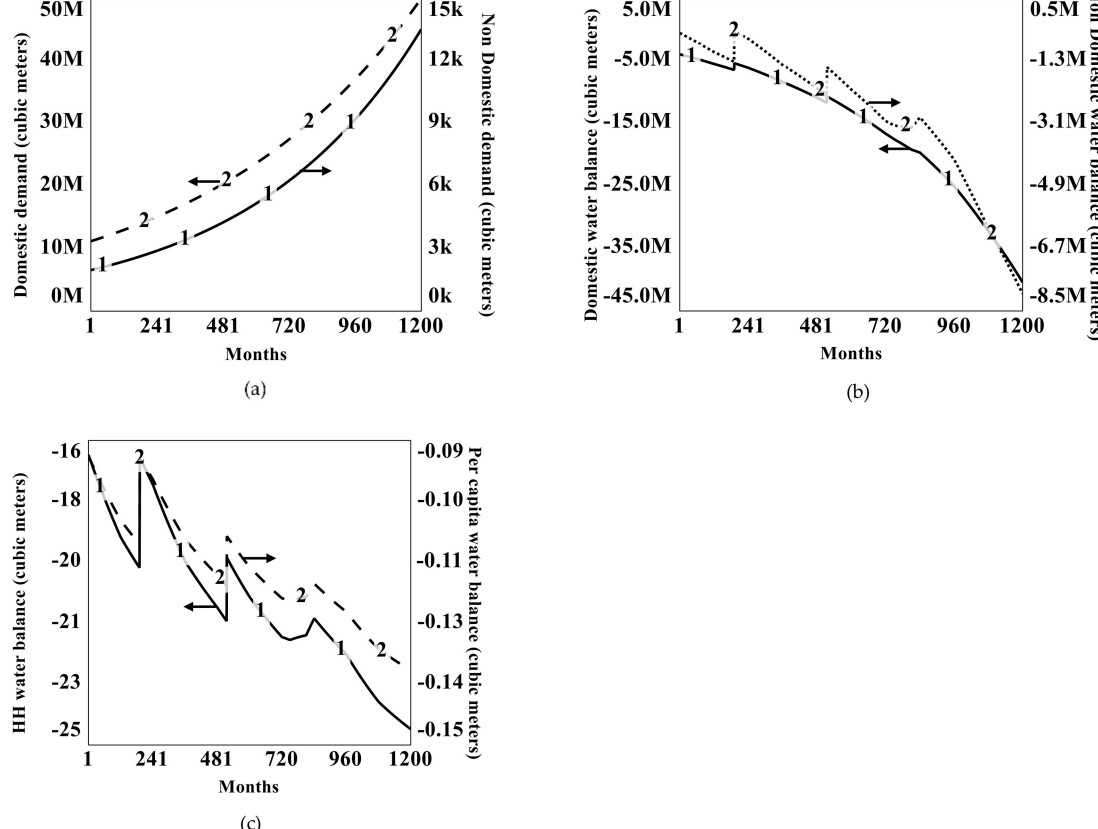

**Figure 2.** (**a**) Total domestic and non-domestic urban water demand, (**b**) total domestic and non-domestic urban water balance and (**c**) water balance at HH and per capita level (increased formal supply capacity of 25MGD and 26 MGD in t = 192 (2023) and t = 516 (2050) decreased the deficit).

### 3.1. Urban Water Demand Growth and Formal Water Sector Balance

Figure 2a demonstrates the total water demand growth, including domestic ($D_D$) and non-domestic ($D_{ND}$) demand categories for the city of Hyderabad. Within the 1.94% population growth rate, domestic water demand outgrew non-domestic water demand in 10 years (t = 120) with a total monthly demand value of 7.95 million cubic meters by December 2017 (t = 120). Additionally, the non-domestic water demand with a growth rate of 1.5% grew up to 6.24 thousand cubic meters per month (Month 120). Further simulations up to 2107 under current population growth rate and non-domestic demand growth rate and domestic water demand indicates a major proportion of the total urban water demand.

Figure 2b displays domestic and non-domestic water balance under the demand growth shown in Figure 2a and future planned changes in supply capacity. Under current distribution proportions of 55:45 for domestic and non-domestic sectors, a high domestic demand deficit can be observed, as the population growth rate is higher than the non-domestic demand. System upgrades at t = 192 and t = 516 decrease the total domestic sector deficit by 16% and 9% and non-domestic sector deficit by 65% and 36%, respectively.

Figure 2c presents formal water systems balance at household and per capita level. The graph shows that by 2030 (t = 274), with current daily supply capacity of 0.2 MCM, water supply from the formal water system in the city drops by 50% of the daily per capita water demand of 0.152 m³. Simulations for 100 years demonstrate that by 2050 the FWSS would no longer be able to meet the minimum per capita demand of 0.050 m³ (50 l), as defined by the World Health Organization (WHO) [43,44]. Hunter, MacDonald and Carter [44] have documented 50 L/c/d as the minimum recommendation while 20 L/c/d has been associated with high risk to health. The WHO suggests optimal access to be equal to or greater than 100 L/c/d. However, infrastructure upgrades (t = 192,

516) and decline in losses due to infrastructure rehabilitation (t = 820) may delay deficits over the simulation period.

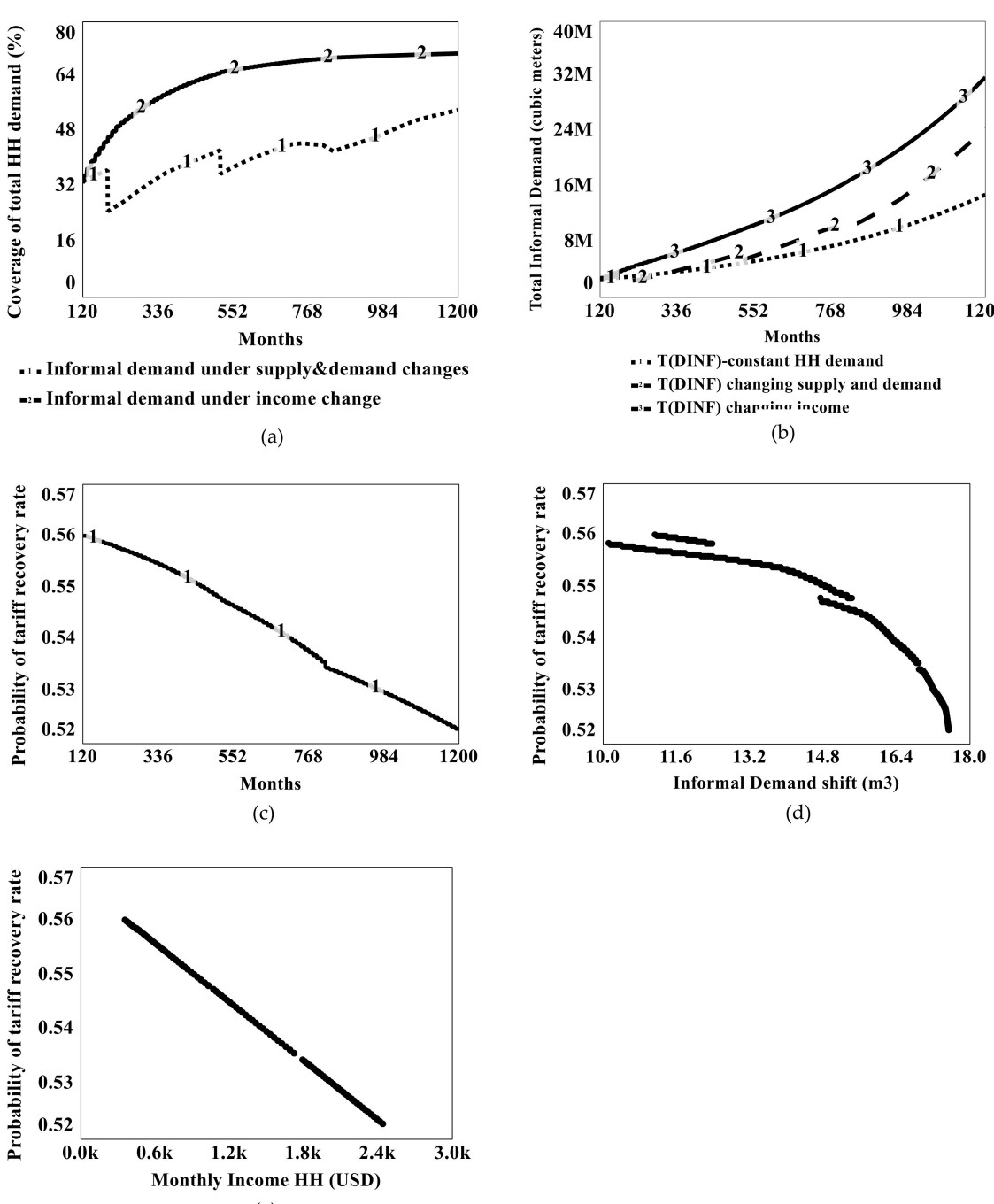

**Figure 3.** (**a**) Informal water demand at HH level under supply demand gap and income change (**b**) Total informal demand (million cubic meters)at city level under current infrastructure capacity, population increase and income change (**c**) Probability of HH's tariff recovery rate over simulation period; (**d**) probability of tariff recovery against informal demand; (**e**) against income change.

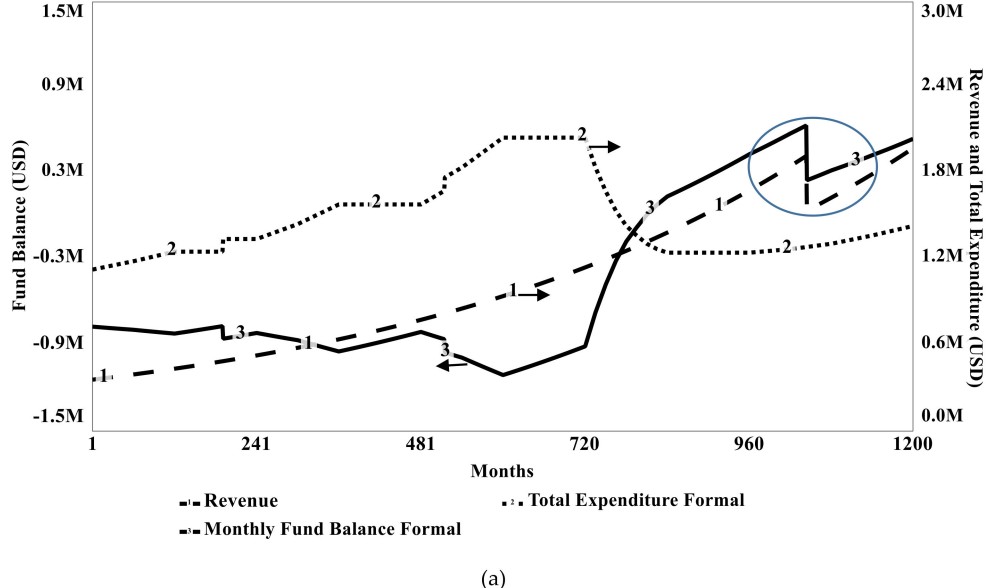

(a)

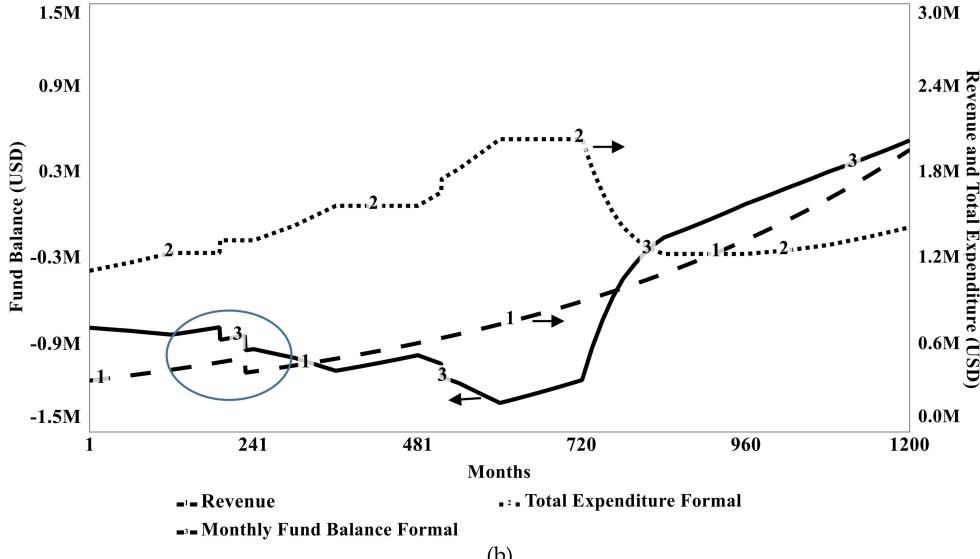

(b)

**Figure 4.** Formal system's fund balance, revenue and total expenditure (Million USD) under (**a**) informal demand shift resulting from demand supply gap and (**b**) informal demand shift due to HH income change. Circled areas present the decline in revenue due to change in recovery rate.

### 3.2. Formal–Informal Demand Shift and the Feedbacks

Figure 3a provides an overall picture of demand shift at the household level from formal to informal against the total HH demand of 27.4 m$^3$/month (the value assumed constant for the whole simulation period). The starting value for the simulation is based on a total informal HH demand value of 33% (9.04 m$^3$) for the year 2016 as observed by Imad [13]. Informal demand increases with a higher rate under the income change scenario, reaching in 50% of the total HH demand by the year 2026 (t = 229). The demand shift is higher throughout the simulation period reaching up to almost 71% of the total demand by the end. The demand shift curve can be observed stable in later years due to lowering income change rate (ranging from 0.4–0.1), representing a higher inflation rate. The informal demand under supply and demand change scenario can be observed to be increasing at a lower rate reaching 54% of the total HH demand by the end of the simulation period. Increase in water treatment

capacity for the years 2025 and 2050, as planned by the Planning and Development Division Pakistan, makes a visible shift in the curve, increasing formal supply and thus reducing the demand shift to informal. The downward shift at t = 840 is associated with the decline in water leakage resulting from infrastructure rehabilitation (as discussed in Section 2.2.2).

Figure 3b provides a comparative analysis of the demand shift trends at the city level presenting the total informal demand growth patterns under constant HH informal demand (9.04 m³), under income change and under supply and demand change scenarios. If the informal demand at the HH level is kept constant, the total demand growth at the city level would only be driven by population growth, reaching a total value of 14.9 MCM by 2107. Total informal demand under income change grows 114% higher (with a value of 31.9 MCM) than the constant demand scenario. Contrastingly, the total demand shift under supply and demand changes grows 65% (25 MCM), higher than the constant HH demand scenario.

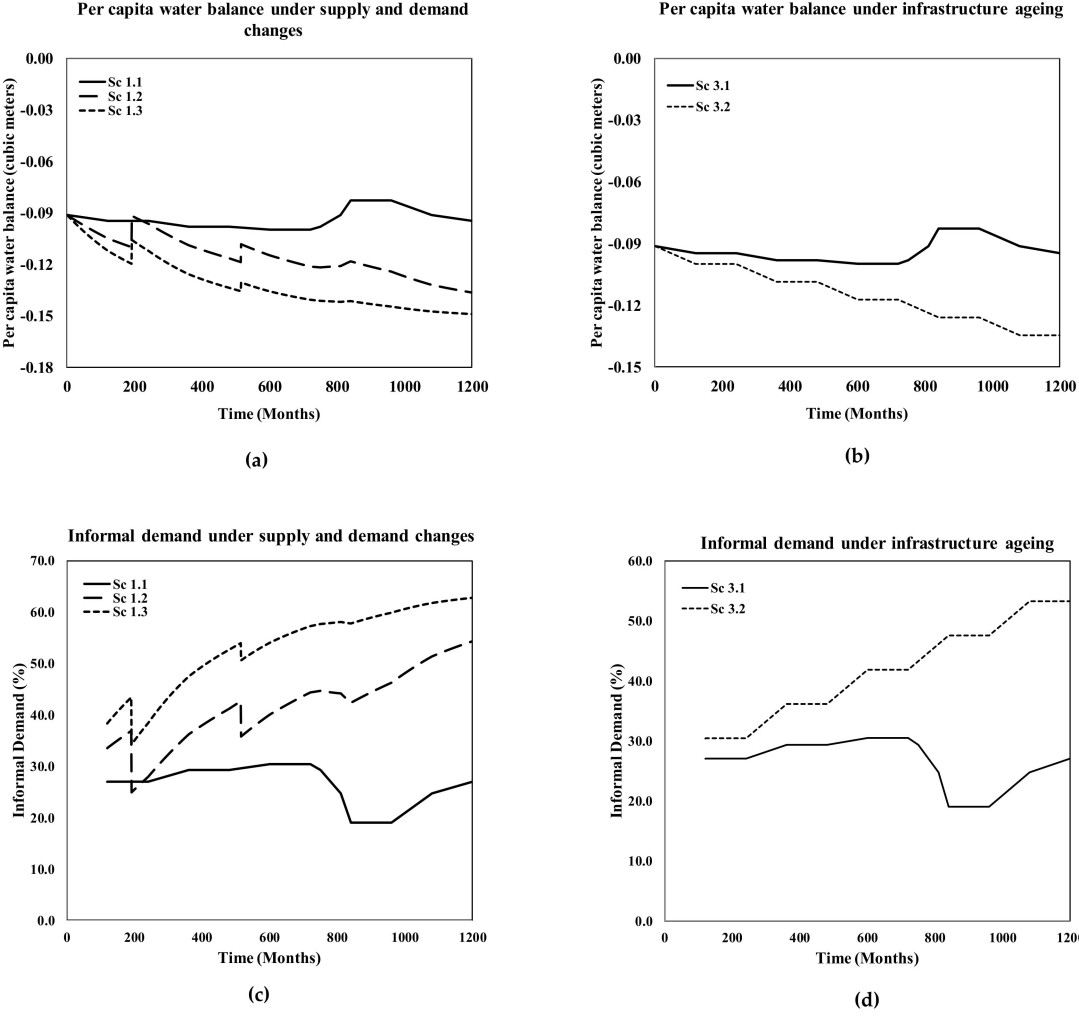

**Figure 5.** Sensitivity analysis results for (**a**) Per capita water balance under supply–demand changes, (**b**) per capita water balance under infrastructure ageing, (**c**) HH demand shift to informal under supply–demand changes, and (**d**) demand shift under infrastructure ageing.

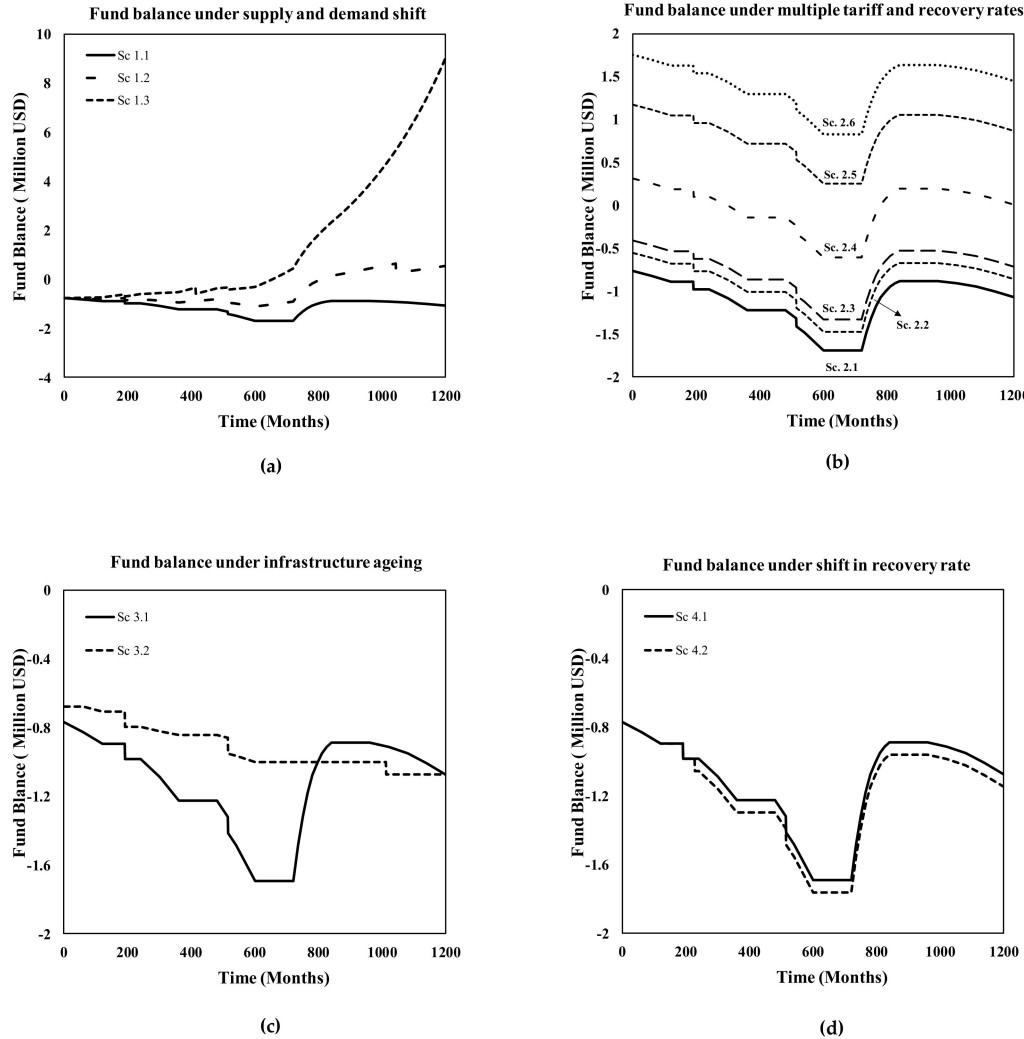

**Figure 6.** Sensitivity analysis results for Formal system fund balance under (**a**) demand supply changes, (**b**) multiple tariff and recovery rates, (**c**) infrastructure ageing, and (**d**) changes in tariff recovery under demand shift to informal.

The regression coefficients mentioned in Equation (8) obtained from logistic regression (provided in Table S4) can further be used to model future trends of probability (p) of $R_T$ at the HH level as shown in Figure 3c–e. Probability of $R_T$ is observed to decline over time with increases in informal demand. However, the change in p($R_T$) is not significant with a 7% decline (56% to 52%) (Figure 3c). Figure 3d,e further display the impacts of informal demand and average income on p($R_T$) individually for the decline in p($R_T$) in Figure 3 is smoother (−2%) until the total informal demand reaches up to 50% (14m$^3$) of the total HH demand (27.4 m$^3$) followed by an abrupt decline in p($R_T$) by 5% for the latter half of the demand value. This decline in p($R_T$) is reflected in the model as a change in recovery rate due to informal demand shift, which determines the *revenue* of the formal system (Equation (11)).

The outcome has also been incorporated as a scenario in sensitivity analysis as displayed in Table 1, allowing $R_T$ to decline by 10% if the informal demand at HH increases above 50% of the total demand.

### 3.3. Fund Balance

The FWSS fund balance analysis (Figure 4) reveals a continuous deficit under increasing *OpEx* with ageing infrastructure and current revenue generation trends. The monthly HH water tariff of

three dollars and the total recovery rate of 50% are too low to generate enough revenue to cover the *OpEx*. The *OpEx* reached up to USD 2.05 million by the service life of the infrastructure (t = 600–721). Infrastructure is assumed to be rehabilitated/replaced after 100 years, resulting in a reduced *OpEx*. However, the *CapEx* of the new or rehabilitated infrastructure has not been incorporated in the model for the reasons explained in Section 2.2.4. *Revenue* is observed to be continuously increasing, primarily driven by population growth. However, it is not enough to balance the *OpEx* until the first 100 years of infrastructure age have elapsed. The impact of demand shift (as discussed in Figure 3b) on recovery rate and the resulting value of *revenue* is observed for supply and demand change scenarios (Figure 4a) and income change (Figure 4b). Informal demand increases above 50% of the total earlier demand under income change, resulting in reduced *revenue* at t = 229. The downward shift of *revenue* under supply and demand change scenario is being observed at time 1045 (Year 2094). Fund balance is observed increase after 100 years of infrastructure age. This happens as a result of decreasing *OpEx* due to rehabilitated infrastructure and increased *revenue* associated with population growth. The impact of demand shift under income change decreases the *revenue* by 20% from t = 229 to 1045.

### 3.4. Sensitivity Analysis

Sensitivity analysis of the per capita formal water balance (as shown in Figure 5a,b) reveals the variable to be more sensitive to supply and demand changes and infrastructure condition. The per capita formal water balance ranged from −0.095 to −0.0151 with the lowest value observed under the highest demand growth scenarios for domestic and non-domestic sectors, and no infrastructure rehabilitation scenario (Sc 1.3), which allowed *water losses* to reach up to 70% of total supply. A higher water balance value was observed under no changes in demand and supply scenario, with infrastructure rehabilitation (Sc 1.1), which allowed the *water losses* to be minimized after 100 years of infrastructure age.

Sensitivity analysis of the informal demand shift (Figure 5c,d) follows the same pattern as that of the per capita water balance with values ranging from 27% to 64% of total HH demand. The demand shift is driven by the demand supply gap where informal demand at the HH level is minimum under lower demand and high supply scenarios. Demand shift would be high both under high population growth rate and infrastructure ageing with no rehabilitation (Figure 5c,d). Sensitivity of the demand shift is not measured against the possible future hydrological changes at Kotri barrage, for the reasons mentioned in Section 2.4. However, assurance of the urban share along with other competing demands such as agriculture may not be possible [45].

The formal system's fund balance values ranged from USD 1.1m to USD 40m (Figure 6). The combination of scenarios resulting in the lowest fund balance included: increasing *OpEx* associated with ageing infrastructure, low monthly fee of USD 3 and a decline in recovery rate associated with increasing informal demand above 50% of the total HH demand and constant supply and demand. Under a constant supply and demand scenario, the population was maintained constant and thus there was no additional *revenue* generated. In contrast, the highest fund balance value was obtained under the combination of scenarios including lowest *OpEx* with minimal infrastructure maintenance and highest *revenue* obtained with higher user fee value of USD 12 and maximum demand growth scenario, which allowed the population of the city to grow at 3.6%. Higher population resulted in higher *revenue* generation. Keeping other variables constant, the fund balance was observed to decrease by 8% under demand shift to informal as driven by income change. Under current demand and supply growth, this decreased by 20% (as observed in Figure 4); thus, the gap may increase under higher demand scenarios.

The reliability of formal system measured in terms of the per capita water balance and the FWSS fund balance revealed that existing infrastructure condition and capacity, along with the population growth, decreased the formal system's ability to meet future municipal demand. This trend is increasingly shifting demand to informal supply (as shown in Figure 5c,d).

## 4. Discussion

Aimed at highlighting how human adaptation approaches feedback and affect the sustenance of formal water supply systems, the study models HH water demand shift and its impact on the formal system's financial balance. A household's "reliance on informal supply" is used as a behavioural response to the formal system's inability to meet the consumers' demand. Income and supply–demand gap under growing demand and deteriorating infrastructure capacity are used to drive the informal growth and indicate a possible shift in demand from formal to informal. The major finding of this study is the behaviour of the probability of $R_T$ to the formal system against the HH demand shift.

Results indicate that the demand shift to informal is influenced more by income than a supply–demand gap (Figure 3a). The decrease in formal demand against increase in income is indicative of the inferior nature of the formal service both in terms of quality and quantity, reflecting system's performance and consumers' low satisfaction with the system [30,46]. Informal demand growth patterns under supply–demand gap were predictably associated with HH water balance as the shift was determined by the demand supply gap (Figure 3) [30]. While it may be argued that the demand shift to informal is mostly a quality issue, as the consumers can deal with the quantity issue by investing in household storage, this may not hold true for all income groups [2,3] and shared living, common in some of the most populous cities like Karachi, Mumbai and Chennai. Thus, water balance and demand shift were sensitive to the same input variables as discussed under sensitivity analysis (Figure 5). Increasing informal demand trends at HH and city level not only reflect higher HH expenditure on water [47], but also increasing future infrastructure investment required to meet the informal demand [3].

It has become recognized over the years that informal supply has emerged as an adaptation strategy to water scarcity (both in terms of quantity and quality) and as a solution to poorly performing formal water supply systems. Further, the poor condition of the water supply and improvement in health records in terms of the prevalence of water-borne diseases has been observed in the study area [48]. This may be attributed to increased access to clean water other than improved health facilities and mass awareness. Thus, the informal system plays a major role in providing improved access to water. However, continuation of the informal supply system as a sole reliable source of improved water may affect lower income populations. An increasing poverty gap in Sindh Pakistan represents the current situation of the lower income group struggling with basic necessities [35,49]. Moreover, although increasing reliance on informal water supply system has increased the system's reliability in terms of increase in access to overall supply; higher cost, quality of water supplied, especially by tankers [30] and disproportional distribution [50] are still major factors that make middle- and lower-income groups pay higher costs or cause them to be excluded from access to safe and affordable water supply for domestic use. These households have limited choice but to depend on unreliable sources. Limited access to water has been determined to cause children to skip school, employees to miss work, domestic violence, and neighbourhood quarrels in Karachi, Pakistan [51].

Results also reveal that consumer behaviour analysis in terms of $p(R_T)$ against the demand shift and income change varied slightly (56% to 52%), i.e., decreased by 7% over the 100-year simulation period (Figure 3c). The negative logistic regression coefficient for demand shift presents a negative relationship between the parameters, however, this shift in demand is not a significant predictor of the $R_T$ to the formal system with a p-value of 0.53. The low significance could be due to several reasons and may require further investigation. For example, logistic regression analysis revealed the mean distribution for the respondents who are paying to the formal system in Hyderabad city to be 0.81, i.e., 81% of the respondents are paying to the formal system. However, WASA (2017) continuously reported a recovery rate to be 50% for the years 2012–2017. Higher reported payments could either be a response bias or because the water fee is too low and affordable for most of the respondents. The sample population of the HH survey, however, was representative of all the economic classes of the residents, covering most of the sub-areas of the city. The area under the ROC curve (AUC) metric value obtained in regression analysis was observed to be 68%, which suggest the model's predictive

performance to be average (Figure S7). Significance of the income in predicting $R_T$ with a p-value of 0.00096, aligns well with economic theory that suggests that income plays a major role in defining the consumers' willingness to pay for improved services that can affect financial sustainability.

The behaviour of p($R_T$) against the demand shift to informal (Figure 3d) suggests that exceeding the informal demand above or equal to 50% of the total HH demand, may trigger a decline in marginal $R_T$. This can be associated with the negative relationship of the financial burden acceptability and water bill burden as discussed by Rehan, Unger, Knight and Haas [19], which may limit the expenditure growth on formal demand. Since the total HH expenditure would increase with an increase in informal demand, consumers may be less willing to pay to the formal system. The decline in recovery rate may, thus, be a major factor in reducing future revenue generation and the formal system's fund balance. Fund balance under demand shift is observed to be declining by 8%, keeping other variables constant (Figure 6d). However, in combination with other factors such as under current demand and supply change (t=144), fund balance would reduce an additional 20% (as observed in Figure 4). Although low *revenue* and higher *OpEx* associated with infrastructure ageing are still major factors affecting fund balance, as also observed by Rehan, Knight, Haas and Unger [8], consideration of other factors such as emerging informal water systems is important in policymaking, as consumer behaviour has remained a major barrier in the financial sustainability of the formal system [6]. The formal system's inability to meet the current water demand continuously affects consumer behaviour. Though affordability is still a major factor especially for lower and middle income class, Imad [13] observed higher willingness to pay (USD 12.14 per month) in these groups, 300% higher than the current value, the Hyderabad city residents are paying to the formal system. The value with more than 50% recovery rate was found to be sufficient to get a positive balance over the first half of the simulation period (Figure 6b). Thus, the lower $R_T$ can be attributed to low satisfaction of the residents with the formal system and availability of the best alternative (informal system).

Low financial sustainability of WASA Hyderabad has been reported to be associated with its low efficiency, supplying poorly treated water [38] and acute water shortages in the city for several days, forcing the poor residents to rely on poor quality groundwater or borrow from neighbours [52]. Moreover, this has also led to non-payments of the salaries to maintenance staff at WASA [52,53] and non-payments of electricity charges and a resulting decline in supply. A power disconnection in 2012 resulted in a short fall of six million gallons, resulting in strikes all over the city [54].

## 5. Conclusions

This study provides a new perspective to the interrelationship of formal and informal supply system in an urban area, which can influence the consumer behaviour, may impact the financial sustainability of the FWSS, and could feedback to affect the reliability of the water supply in the long run. The formal system of Hyderabad city does not have the capacity to meet the urban demands and is continuously failing with poor financial structure. While no significant decline in the Indus River flow has been observed in past flow trends, the FWSS does not have the financial capacity to meet the pumping and treatment cost to supply safe drinking water.

Human feedback has been reflected through consumers' behaviour in terms of demand shift and tariff recovery rate. Although, an increasing shift to informal, driven by supply demand gap and income change, has been observed, its impact on the financial balance of the formal system is minimal as the tariff rates are too low for the city to generate enough revenue. However, this could also be because the tariff value was assumed constant for the simulation period.

The water supply system of Hyderabad, Pakistan represents the condition of water services in most of the global south. Formal and informal supply systems together make the city's waterscape. Reliance on informal water supply has been observed throughout and formal systems are struggling with financial sustainability. Effective planning thus requires incorporation of the INFWSS in future planning and policy. Modelling of all the complexities involved in Hyderabad's water supply system is beyond the scope of this paper. There are multiple factors which allow and promote the access to the



informal water supply in the city including, continuous failure of the FWSS in cities like Hyderabad and requires an understanding of the evolving consumer behaviour which may feed back to the system failure. The emergence of the INFWSS as an adaptive approach may be beneficial in the short run but may drive the urban water supply system to be unreliable in the long run. Understanding of the socio-hydrological processes is thus necessary for identifying the factors which can affect the FWSS.

To simplify the model, no climate-related changes in hydrology were considered. A city like Hyderabad with a single source of supply, i.e., surface water, needs incorporation of climate change factors to know the future uncertainties. Future studies which incorporate this aspect could be valuable for a more sophisticated socio-hydrological model.

**Supplementary Materials:** The following are available online at http://www.mdpi.com/2073-4441/12/10/2795/s1, Figure S1: System components of the socio-hydrological processes captured in the study, Figure S2: Formal water balance sector, Figure S3: Infrastructure condition sector, Figure S4: Consumer sector, Figure S5: Financial balance sector. Figure S6: Infrastructure ageing, Figure S7: Informal demand versus formal supply balance per household, Figure S8: The Area under the ROC Curve - Logistic Regression analysis, Text S1: Informal demand and WTP to formal (Logistic Regression Analysis), Table S1: Logistic Regression Coefficients and Table S2: Calibrated input data values. Supplementary material provides details of the System Dynamics (SD) model sectors representing the relationships between input and output variables and preliminary data analysis completed to obtain the logistic regression coefficients to be used in the SD model. Datasets used in the study including: 'Upstream water flow at Kotri barrage, Sindh Pakistan' and 'Household's willingness (WTP) to pay to the utility' are available at Edith Cowan University's data repository and can be accessed at http://dx.doi.org/10.25958/5efd7159d64fb and http://dx.doi.org/10.25958/5efd3803d64fa.

**Author Contributions:** Conceptualization, methodology and analysis and drafting, R.B., supervision M.K., review and editing, M.K. and S.B. All authors have read and agreed to the published version of the manuscript.

**Funding:** The financial support for this research is provided by Edith-Cowan University and Higher Education Commission of Pakistan (ECU-HEC) graduate scholarship.

**Acknowledgments:** We acknowledge USPCASW, Mehran University, WASA Hyderabad and Usama Imad for sharing baseline data.

**Conflicts of Interest:** The author declares no conflict of interest.

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
