# Peer review of "Socio-Hydrological Modelling to Assess Reliability of an Urban Water System under Formal-Informal Supply Dynamics"

_water, doi:10.3390/w12102795_

Round 1
Reviewer 1 Report
The study, presenting a socio-hydrological modelling to assess reliability of an Urban Water System under Formal-Informal Supply Dynamics, is an interesting manuscript. It is well written and organised, as well as the structure balanced. The English languange and style needs to be minor revised. Below some suggestions to improve the quality of the work.
The authors should increase the reference list, by adding more recently studies on the field. They also have to highiglith better the novelty of the study and ow it can overcome the scientific gap in the field.
Section 2 should be called "Methodology" , while "Modelling Approach" can be a sub-section.
The reference riported in line 94 needs to be changed in the correct journal format.
In 2.5 the sensitivity analysis was introduced. Anyway, the authors should improve this sub-section by adding more information about the sensitivy analysis method selected to carried out the analysis. They could also refer to other studies in the field.
Please check also the affiliation of the authors, "4" is missing.
In conclusion, if the authors will address this few issues, the paper can be revised once again before to be approved for publication.
Reviewer 2 Report
09.11.20
Review of: Socio-hydrological Modelling to Assess Reliability of an Urban Water System under Formal-Informal Supply Dynamics.
Draft manuscript submitted to Water.
General Comments: This draft manuscript portends to examine the social-hydrological dynamics of the relationship between informal and formal water supply “deliveries” in Hyderabad Pakistan. Reasonably well written, this draft can contribute to the current body of literature. However, the document needs much work to be accepted for publication.
- You need to reformat your figures; they do not “work” as is. A) Embedded text font is too small, B) The legends do not sufficiently illustrate the specific curves that you are referencing, C) Axes titles on some of your figures bleed into the axis values, D) Figures need to be “stand-alone.” That is, a reader should be able to understand the figure from the symbols and text provided within the figure.
- You use both present and past tense in your paragraphs. You cannot do this.
- In many cases it appears that you could combine small paragraphs. You have far too many single and two or three sentence paragraphs. A one to three sentence paragraph is NOT a paragraph.
- While the authors are fairly proficient in English, and the writing style nearly conforms to standard traditions, the writing could be tightened up by:
- Organizing the paragraphs in each section in a logical, consistent manner
- Keep consistent tense within a paragraph. One typically uses past tense because the study has already been conducted. It is in the past.
- You would be better served by using stronger verbs in your writing. You can address this by using an active rather than a passive voice.
- Ensure that each paragraph has:
- An opening sentence; one that tells the reader what they will find in the paragraph. It sets up the stage for the arguments presented.
- A body of the sentence that addresses the elements in the opening sentence.
- A concluding sentence, or one that leads the reader onto a new thought that is addressed in the following paragraph.
- Watch (look for) redundancy. Your draft has several places where you are reduntant.
Specific Comments:
Line 40. You already mentioned informal water supply systems in your first paragraph. Move the definition up to where you first use it. Remove the redundancy.
Line 56. I doubt that we “understand” the coupled human-water systems. We have a better understanding, perhaps.
Line 57: This sentence uses the pronoun “it” without clear reference to the noun it has replaced. The subject of your first sentence is “literature.” Thus, your pronoun refers to literature… and not socio-hydrology. Thus, the sentence does not read as intended.
Line 67. This is an awkward paragraph. Re-write it.
Line 84. You use “firstly” without a “secondly;” Awkward.
Line 87. “A sensitivity analysis ..” or, Sensitivity analyses are ….
Line 90 is redundant. You mention this on line 82.
Line 141. One sentence does not a paragraph make. Create a topic sentence for a paragraph and then combine a few of these one and two sentence paragraphs.
Line 173. On the contrary, water loses recharge the aquifer. Even if the aquifer water is too brackish to use. Losses are not of no use. However, they are not usable for potable drinking water because they are lost from the infrastructure system.
Line 222. 10m3 over what time period? Monthly? HH- is this not water use? Or, water demand? So, “balance” is not the accurate term here.
Line 328. What approach did you use for your sensitivity analyses? Silva et al. (2007) is one approach. Salelli et al. (e.g., 2002) is another. I see no mention of how you conducted your sensitivity analysis.
Line 334. Table 1 needs work (on the formatting).
Line 365: Figure 2 needs reformatting; it does not work as is.
Line 366. Your figure caption lists years but there are no years in your figure, so that information is meaningless. Use months, or change your figure.
Line 521. Your “major” finding is found in the seventh paragraph of your discussion. ????
Discussion section: I suspect that you could create appropriate paragraph sentences to combine some of the current paragraphs that you have in the discussion section. Overall the discussion section is not very well organized, as is.
Round 2
Reviewer 2 Report
Much improved.